# Learning Equivariant Segmentation with Instance-Unique Querying

**Wenguan Wang**[*]
ReLER, AAII, University of Technology Sydney

**James Liang**[*]
Rochester Institute of Technology

**Dongfang Liu**[†]
Rochester Institute of Technology

## Abstract

Prevalent state-of-the-art instance segmentation methods fall into a *query*-based scheme, in which instance masks are derived by querying the image feature using a set of instance-aware embeddings. In this work, we devise a new training framework that boosts query-based models through discriminative query embedding learning. It explores two essential properties, namely *dataset-level uniqueness* and *transformation equivariance*, of the relation between queries and instances. First, our algorithm uses the queries to retrieve the corresponding instances from the whole training dataset, instead of only searching within individual scenes. As querying instances across scenes is more challenging, the segmenters are forced to learn more discriminative queries for effective instance separation. Second, our algorithm encourages both image (instance) representations and queries to be equivariant against geometric transformations, leading to more robust, instance-query matching. On top of four famous, query-based models (*i.e.*, CondInst, SOLOv2, SOTR, and Mask2Former), our training algorithm provides significant performance gains (*e.g.*, $+1.6 - 3.2\,AP$) on COCO dataset. In addition, our algorithm promotes the performance of SOLOv2 by 2.7 *AP*, on LVISv1 dataset.

## 1 Introduction

Instance segmentation, *i.e.*, labeling image pixels with classes and instances, plays a critical role in a wide range of applications, *e.g.*, autonomous driving, medical health, and augmented reality. Modern instance segmentation solutions are largely built upon three paradigms: *top-down* ('detect-then-segment') [1–19], *bottom-up* ('label-then-cluster') [20–28], and *single-shot* ('directly-predict') [29–44]. Among them, the top-leading algorithms [18,34,39–44] typically operate in a *query*-based mode, in which a set of instance-aware embeddings is learned and used to query the dense image feature for instance mask prediction. The key to their triumph is the instance-aware query vectors that are learned to encode the characteristics (*e.g.*, location, appearance) of instances [34,43]. By straightforwardly minimizing the differences between the retrieved and groundtruth instance masks, the query-based methods, in essence, learn the query vectors for instance discrimination only *within individual scenes*.

As a result, existing query-based instance segmentation algorithms place a premium on intra-scene analysis during network training. Since the scenario in one single training scene is simple, *i.e.*, the diversity and volume of object instances as well as the complexity of the background are typically limited, learning to distinguish between object instances only within the same training scenes is less challenging, and inevitably hinders the discrimination potential of the learned instance queries.

---

[*]authors contributed equally

[†]corresponding author

36th Conference on Neural Information Processing Systems (NeurIPS 2022).

This work brings a paradigm shift in training query-based instance segmenters: it goes beyond the *de facto*, within-scene training strategy by further considering the cross-scene level query embedding separation of different instances – *querying instances from the whole training dataset*. The underlying rationale is intuitive yet powerful: an advanced instance segmenter should be able to differentiate all the instances of the entire dataset, rather than only the ones within single scenes. Concretely, in our training framework, the queries are not only learned to fire on the pixels of their counterpart instances in the current training image, but also forced to mismatch the pixels in other training images. By virtue of intra-and inter-scene instance disambiguation, our framework forces the query-based segmenters to learn more discriminative query vectors capable of uniquely identifying the corresponding instances even at the dataset level. To further facilitate the establishment of robust, one-to-one relation between queries and instances, we complement our training framework with a *transformation equivariance* constraint, accommodating the equivariance property of the instance segmentation task to geometric transformations. For example, if we crop or flip the input image, we expect the image (instance) features and query embeddings to change accordingly, so as to appropriately reflect the variation of instance patterns (*e.g.*, scale, position, shape, *etc*) caused by the input transformation.

Exploring intra-and inter-scene instance uniqueness as well as transformation equivariance leads to a general yet powerful training framework. Our algorithm, in principle, can be seamlessly incorporated into the training process of existing query-based instance segmenters. For comprehensive evaluation, we apply our algorithm to four representative, query-based models (*i.e.*, CondInst [34], SOLOv2 [39], SOTR [41], and Mask2Former [40]) with various backbones (*i.e.*, ResNet [45], Swin [46]). Experiments on COCO [47] verify our impressive performance, *i.e.*, **+2.8 – 3.1**, **+2.9 – 3.2**, **+2.4 – 2.6**, and **+1.6 – 2.4** AP gains over CondInst, SOLOv2, SOTR, and Mask2Former, respectively (see Fig. 1). Our algorithm also brings remarkable improvement, **+2.7 AP**, on LVISv1 [48] dataset, on top of SOLOv2 [39]. These results are particularly impressive considering our training algorithm causes neither architectural change nor extra computational load during model deployment.

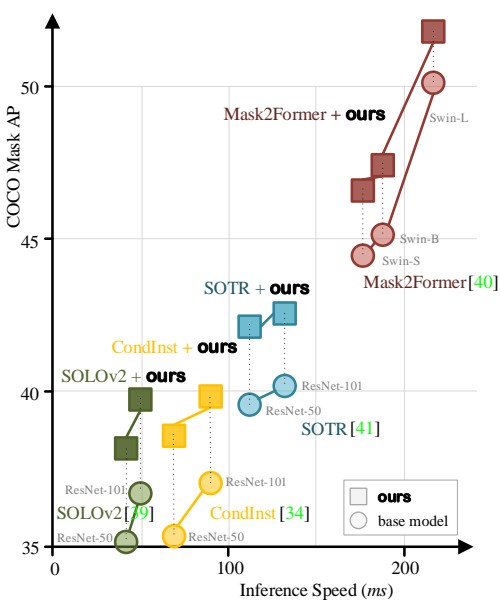

Figure 1: Our training algorithm yields solid performance gains over state-of-the-art query-based models [34, 39–41] without architectural modification and inference speed delay.

## 2 Related Work

This section summarizes the most relevant research on instance segmentation and equivariant learning.

**Instance Segmentation.** With the renaissance of connectionism, remarkable progress has been made in instance segmentation. Existing deep learning based solutions can be broadly classified into three paradigms: top-down, bottom-up, and single-shot. Following the idea of '*detect-then-segment*', **top-down** methods [1–18] predict a bounding box for each object and then ouput an instance mask for each box. Though effective, this type of methods is complicated and dependent on the priori detection results. In contrast, **bottom-up** methods [20–27] adopt a '*label-then-cluster*' strategy: learning per-pixel embeddings and then grouping them into different instances. Albeit simple, this type of methods relies on the performance of post-processing and easily suffers from under-segment or over-segment problems. Inspired by the advance of single-stage object detection [49,50], a few recent efforts approach instance segmentation in a **single-shot** fashion, by coalescing detection and segmentation over pre-defined anchor boxes [29–33], or directly predicting instance masks from feature maps [34–44,51]. This type of methods is well recognized and generally demonstrates better speed-accuracy trade-off [52].

Despite the blossoming of diverse approaches, the vast majority of recent top-performing algorithms [18, 34, 39–44] fall into one grand category – *query*-based models. The query-based methods utilize compact, learnable embedding vectors to represent instances of interest and leverage them as queries to decode masks from image features. Their triumph is founded on comprehensively encoding instance-specific properties (*e.g.*, location, appearance) into the query vectors, which significantly increases prediction robustness. For instance, [34, 39] exploit the technique of dynamic filter [53] to

generate instance-specific descriptors, which are convolved with image feature maps for instance mask decoding. Inspired by DETR [54], [40, 42, 43] alternatively leverage a Transformer decoder to obtain instance-aware query embeddings and cast instance segmentation as a set prediction problem.

Our contribution is orthogonal and these query-based segmenters can benefit. We scaffold a new training framework that sharpens the instance discriminative capability of the query-based segmenters. This is achieved by matching query embeddings with instances within and cross training scenes. Such intra-and inter-scene instance querying strategy is further enhanced by an equivariance regularisation term, addressing not only the uniqueness but also the robustness of instance-query relations.

**Equivariant Representation Learning.** Transformations play a critical role in learning expressive representations by transforming images as a means to reveal the intrinsic patterns from transformed visual structures [55]. Motivated by the concept of *translation equivariance* underlying the success of CNNs, numerous efforts (*e.g.*, capsule nets [56,57], group equivariant convolutions [58], and harmonic networks [59]) investigate learning more powerful representations *equivariant* to generic types of transformations [60, 61]. A representation $f$ is said to be equivariant with a transformation $g$ for input (say image) $I$ if $f(g(I)) \approx g(f(I))$. In other words, the output representation $f(I)$ transforms in the same manner (or, in a broad sense, a predictable manner) given the input transformation $g$. Many recent self-supervised learning methods [62–64] encourage the representations to be *invariant* under transformations, *i.e.*, $f(g(I)) \approx f(I)$. As such, invariance can be viewed as a special case of equivariance [65] where the output representation $f(I)$ does not vary with the input transformation $g$.

In our training framework, we fully explore the inherent, transformation-equivariance nature of the instance segmentation task to pursue reliable, one-to-one correspondence between learnable queries and object instances. This is accomplished by promoting the equivariance of both query embeddings and feature representations with respect to spatial transformations, *i.e.*, cropping or flipping of an input image should result in correspondingly changed feature representation, query embeddings, as well as instance mask predictions. Note that invariance is not suitable for instance segmentation task, as it encourages the feature map (and segmentation mask) to not vary with the input transformation. Our algorithm is also in contrast to the common data augmentation strategy, in which the transformed images and annotations are used directly as additional individual training examples, without any constraint about the relation between the representations (and queries) produced from the original and transformed views. Our experimental results (see §4.3) also evidence the superiority of our transformation equivariance learning over transformation-based data augmentation.

## 3 Methodology

Next, we first formulate instance segmentation from a classical view of mask prediction and classification (§3.1). Then we describe our new training framework (§3.2) and implementation details (§3.3).

### 3.1 Problem Formulation

Instance segmentation seeks a partition of an observed image $I \in \mathbb{R}^{H \times W \times 3}$ into $K$ instance regions:

$$\{Y_k\}_{k=1}^K = \{(M_k, c_k)\}_{k=1}^K, \quad \text{where} \ \ M_k \in \{0,1\}^{H \times W}, \ \ c_k \in \{1, \cdots, C\}. \tag{1}$$

Here the instances of interest are represented by a total of $K$ *non-overlap*, binary masks $\{M_k\}_{k=1}^K$ as well as corresponding class labels $\{c_k\}_{k=1}^K$ (*e.g.*, table, chair, *etc*). For a pixel $i \in I$, its counterpart value in the *k-th* groundtruth mask, *i.e.*, $M_k(i)$, denotes whether $i$ belongs to instance $k$ (1) or not (0). Note that the number of instances, $K$, varies across different images. Existing mainstream (or, more precisely, most top-down and one-shot) solutions approach the task by decomposing the image $I$ into a fixed-size set of soft masks. In this setting, each mask is associated with a probability distribution over all the $C$ categories. The output can thus be represented as a set of $N$ mask-probability pairs:

$$\{\hat{Y}_n\}_{n=1}^N = \{(\hat{M}_n, \hat{p}_n)\}_{n=1}^N, \quad \text{where} \ \ \hat{M}_n \in [0,1]^{H \times W}, \ \ \hat{p}_n \in \triangle^C. \tag{2}$$

Here $\triangle^C$ stands for the $C$-dimensional probability simplex. The size of the prediction set, $N$, is usually set as a constant and much larger than the typical number of object instances in an image. Hence the training objective penalizes the errors of both label prediction and mask estimation:

$$\mathcal{L}(\{\hat{Y}_n\}_{n=1}^N, \{Y_k\}_{k=1}^K) = \sum_{n=1}^N \mathcal{L}_{cls}(\hat{p}_n, c_{\sigma(n)}) + \mathcal{L}_{mask}(\hat{M}_n, M_{\sigma(n)}), \tag{3}$$

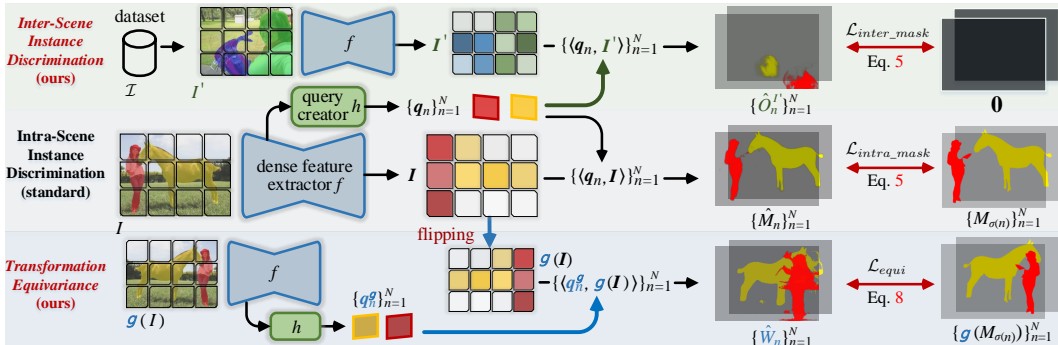

Figure 2: Overview of our new training framework for query-based instance segmentation. Rather than current *intra-scene* training paradigm, our framework addresses *inter-scene instance discrimination* and *transformation equivariance* for discriminative instance query embedding learning (see §3.2). To improve readability, for image $I$, we only plot one extra image $I'$ for cross-scene training.

where $\sigma$ refers to the matching between the prediction and groundtruth sets (established by certain rules [6, 54]). The classification loss $\mathcal{L}_{cls}$ is typically the cross-entropy loss or focal loss [66]; the mask prediction loss $\mathcal{L}_{mask}$ can be the cross-entropy loss in [40], dice loss [67] in [34, 39], or focal loss in [54]. While many approaches supplement $\mathcal{L}$ with various extra losses (*e.g.*, bounding box loss [6, 32, 34, 54, 68], ranking loss [69], semantic segmentation loss [13, 32]), later we will show our *cross-scene* training scheme is fundamentally different from (yet complementary to) current *scene-wise* training paradigm.

## 3.2 Equivariant Learning with Intra-and Inter-Scene Instance Uniqueness

**Query-based Instance Segmentation.** As clearly indicated by Eq. 2, the prediction masks $\{\hat{M}_n\}_{n=1}^N$ are the means of separating instances at the pixel level. Current top-performing instance segmenters [18, 34, 39–43] typically generate mask predictions in a *query*-based fashion (see the middle part of Fig. 2). Let $f$ be a *dense feature extractor* 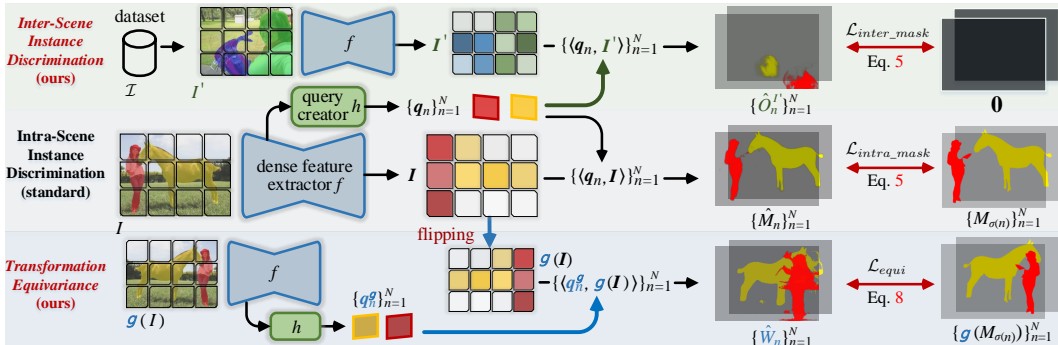 (*e.g.*, an encoder-decoder fully convolutional network [70]) that produces $D$-dimensional dense embedding $\boldsymbol{I}$ for image $I$, *i.e.*, $\boldsymbol{I} = f(I) \in \mathbb{R}^{H \times W \times D}$. Then a *query creator* $h$ is adopted to produce a set of $N$ instance-aware embedding vectors $\{\boldsymbol{q}_n \in \mathbb{R}^d\}_{n=1}^N$, which are used to query the image representation $\boldsymbol{I}$ for instance mask decoding:

$$\{\hat{M}_n\}_{n=1}^N = \{\langle \boldsymbol{q}_n, \boldsymbol{I} \rangle\}_{n=1}^N, \quad \text{where} \ \{\boldsymbol{q}_n\}_{n=1}^N = h(\boldsymbol{I}^{\Downarrow}). \tag{4}$$

Here $\langle \cdot, \cdot \rangle$ is a certain similarity measure performed pixel-wise, and $\boldsymbol{I}^{\Downarrow}$ (typically) refers to a low-resolution feature representation of image $I$. Note that position information is integrated to either or both of $\boldsymbol{I}$ and $\boldsymbol{I}^{\Downarrow}$, to make the model *location-sensitive*. The query creator $h$ is implemented as a dynamic network [34, 39], or a Transformer decoder [40, 42]. For dynamic network based $h$, it predicts $N$ convolution filters $\{\boldsymbol{q}_n\}_{n=1}^N$ dynamically conditioned on the input $\boldsymbol{I}^{\Downarrow}$, and hence $\langle \cdot, \cdot \rangle$ refers to convolution ($d \neq D$). For Transformer decoder based $h$, it additionally leverages a set of $N$ learnable positional embeddings (omitted for brevity) to gather instance-related context from $\boldsymbol{I}^{\Downarrow}$; the collected context is stored in $\{\boldsymbol{q}_n\}_{n=1}^N$ and $\langle \cdot, \cdot \rangle$ is computed as dot product for instance mask decoding ($d = D$). Eq. 4 informs that, query-based segmenters in essence learn $N$ compact descriptors $\{\boldsymbol{q}_n\}_{n=1}^N$ to grasp critical characteristics (*e.g.*, appearance and location) of potentially interested instances, and use these instance descriptors as queries to retrieve corresponding pixels from $\boldsymbol{I}$. It is thus reasonable to assume the discriminative ability of the learned query embeddings is crucial for the performance of query-based methods. Viewed in this light, a question naturally arises: ❷ *how to learn discriminative instance query embeddings*? Yet, this fundamental question is largely ignored in the literature so far.

To respond ❷, we exploit two crucial properties of instance-query matching, namely *uniqueness* and *robustness*. This is achieved by addressing dataset-level uniqueness and transformation equivariance during the learning of query-based segmenters, leading to a powerful training scheme eventually.

**Learning with Intra-and Inter-Scene Instance Uniqueness.** If we closely scrutinize at the current *de facto* training regime (*cf.* Eq. 3) and the work mode of query-based segmenters (*cf.* Eq. 4), we can find: the mask prediction loss $\mathcal{L}_{mask}$ forces each query $\boldsymbol{q}_n$ to match the pixels $i \in I$ of its counterpart instance $k = \sigma(n)$, *i.e.*, $M_{\sigma(n)}(i) = 1$, and mismatch the pixels $i' \in I$ of other instances $k \neq \sigma(n)$, *i.e.*,

$M_{\sigma(n)}(i') = 0$, *within* image $I$. That means, the segmenters are only taught to differentiate instances within *individual* scenes. Evidently, individual scenes are unable to cover diverse instance patterns; the scarcity of negative instances — an intrinsic limitation of the current *scene-wise* training regime — inevitably impairs query embedding learning. This analysis motivates us to design a *cross-scene* training strategy, where the segmenters are forced to learn to query the target instances from the whole training dataset, instead of only matching queries and instance pixels within the same scenes. Thus we take a step further by simultaneously leveraging the intra-and inter-scene context to promote query embedding learning. Query-based segmenters are encouraged to develop dataset-level, cross-instance discriminativeness and perform reliable mask prediction using instance uniqueness.

Let $\mathcal{I}$ be the training set of $|\mathcal{I}|$ images and $\{q_n\}_{n=1}^N$ the instance queries generated from image $I \in \mathcal{I}$. During training, rather than only applying $\{q_n\}_{n=1}^N$ for the counterpart image $I$ to get $N$ *intra-image* instance prediction masks $\{\hat{M}_n\}_{n=1}^N$ (*cf.* Eq. 4), we further use $\{q_n\}_{n=1}^N$ to query other training images $I' \in \mathcal{I}$ for cross-scene instance disambiguation. This will result in $N$ *inter-image* instance prediction masks $\{\hat{O}_n^{I'} \in [0,1]^{H \times W}\}_{n=1}^N$ for each $I'$, *i.e.*, $\{\hat{O}_n^{I'}\}_{n=1}^N = \{\langle q_n, \boldsymbol{I'} \rangle\}_{n=1}^N$, where $\boldsymbol{I'} \in \mathbb{R}^{H \times W \times D}$ is the feature map of $I'$, *i.e.*, $\boldsymbol{I'} = f(I')$. For each query $q_n$ of image $I$, given its corresponding intra-image instance prediction map $\hat{M}_n$, a total of $|\mathcal{I}| - 1$ inter-image instance prediction maps $\{\hat{O}_n^{I'}\}_{I' \neq I}$, and the original groundtruth mask $M_{\sigma(n)}$, we improve the mask prediction loss $\mathcal{L}_{mask}$ in Eq. 3 as a form of:

$$\sum\nolimits_{n=1}^N \left( \mathcal{L}_{intra\_mask}(\hat{M}_n, M_{\sigma(n)}) + \frac{1}{|\mathcal{I}| - 1} \sum\nolimits_{I' \neq I} \mathcal{L}_{inter\_mask}(\hat{O}_n^{I'}, \mathbf{0}) \right), \qquad (5)$$

where the first term $\mathcal{L}_{intra\_mask}$ is exactly the same as $\mathcal{L}_{mask}$ in Eq. 3, which is renamed for addressing its nature of inspiring intra-scene instance uniqueness. The newly added term $\mathcal{L}_{inter\_mask}$ is to eliminate the instance ambiguity through inter-scene querying. Thus its training target is an all-zero query-pixel matching matrix $\mathbf{0}$ of size $H \times W$, *i.e.*, each query $q_n$ of image $I$ is forced to mismatch the instances in other images $I'$. Among the several choices for $\mathcal{L}_{inter\_mask}$, *e.g.*, $\ell_1$ loss, $\ell_2$ loss, and cross-entropy loss, we find focal loss [66] is more favored (*cf.* §4.3). We speculate this is due to the advantage of focal loss in preventing the influence of massive trivial examples, as in our case. To further boost the effectiveness and efficiency of our algorithm, the following strategies are adopted:

- *External memory*: As the capacity of training batch is limited, we follow the recent practice in unsupervised representation learning [62, 71–75] to build an external memory to enable large-scale query-instance matching. The memory gathers instance pixel embeddings from several batches. Note that we drop background pixels, as we find their contribution is negligible (sometimes even negative). This also implies that cross-instance discrimination is the core challenge of this task.
- *Sparse sampling*: As local image contents are highly correlated, storing all the pixel samples will bring a lot of redundant information, which is useless for discriminative representation learning. With a similar spirit of some dense prediction methods [15, 73], only a small set of instance pixels are randomly sampled from each image and fed into the memory. In practice, we find this strategy greatly improves the diversity of the instance samples and eventually benefits the final performance.
- *Instance-balanced sampling*: Though effective, the sparse sampling strategy has a side-effect: it hurts the performance on small instances. This is because larger instances have more chance of being sampled for loss computation, enabling our training algorithm to give less priority towards smaller instance. We therefore opt to randomly sample a fixed number of pixels from each instance region. As a result, the performance is further improved. All the related experiments can be found in §4.3.

These designs together form a cross-scene training framework, whose power lies in the query embedding uniqueness among plentiful instances. Our framework is elegant and principled: as suggested by Eq. 5, it can complement and integrate with current training paradigm of query-based methods.

**Learning with Transformation Equivariance.** So far, we have addressed ❷ from a fresh perspective of dataset-level instance-unique querying. Next, we complete our response from another essential viewpoint: *robustness*. That is, we expect the segmenters to build robust instance-query correspondences that are equivariant against input imagery transformations (*e.g.*, cropping, flipping, *etc*).

Our key insight for robustness is derived from the transformation equivariance nature of the task: flipping or cropping the image should result in an exact change in the instance segmentation mask. At the first glimpse, this character seems to have been already captured by the widely-used, transformation-based data augmentation technique. Under scrutiny, this view however is erroneous because: the representations (and instance queries) ought to be transformation equivariant first, then it is natural to achieve correctly transformed predictions. Yet, the current training strategy puts the cart before

the horses: with input transformations, the segmenter is only trained to produce correspondingly transformed predictions, but without any reasonable constraints on the representations (and queries).

Given the analysis above, we supplement our framework with an equivariance based training objective, encouraging the segmenters to precisely and reliably anchor queries on instances. Let $G$ be a group of transformations (*e.g.*, cropping, flipping, *etc*). For a query-based instance segmenter, we encourage its feature representation to be "equivariant against input imagery transformations", which states:

$$\forall g \in G : f(g(I)) \approx g(f(I)) = g(\boldsymbol{I}). \tag{6}$$

Intuitively, the output representation of $f$ is encouraged to change in the same way to the transformation $g$ applied to the input $I$. Similarly, the instance queries are also desired to be transformation equivariant, *i.e.*, they should be able to properly describe the transformation applied to their counterpart instances. For example, after cropping or flipping an input image, the queries are expected to still robustly capture the changed instance patterns (*e.g.*, shape, location, scale). We formulate this property as:

$$\forall g \in G : \{\langle \boldsymbol{q}_n^g, \boldsymbol{I}^g \rangle\}_{n=1}^N \approx \{g(M_{\sigma(n)})\}_{n=1}^N, \tag{7}$$

where $\boldsymbol{I}^g$ denotes the feature embedding of transformed image (instances) $g(I)$, *i.e.*, $\boldsymbol{I}^g = f(g(I))$; analogously, $\{\boldsymbol{q}_n^g\}_{n=1}^N$ indicates the query embeddings derived from $g(I)$. Eq. 7 states that, with a transformed input image $g(I)$, the queries should be properly changed so as to correctly match the embeddings of transformed instances $\boldsymbol{I}^g$. This also explains the rationale behind the popular transformation-based augmentation technique from a view of equivariant query embedding learning.

Considering both Eq. 6 and 7, we can easily have $\forall g \in G : \{\langle \boldsymbol{q}_n^g, g(\boldsymbol{I}) \rangle\}_{n=1}^N \approx \{g(M_{\sigma(n)})\}_{n=1}^N$, and hence our equivariance based training objective is formulated as (see the bottom part of Fig. 2):

$$\sum_{n=1}^N \mathcal{L}_{equi}(\hat{W}_n^g, g(M_{\sigma(n)})), \quad \text{where } \hat{W}_n^g = \langle \boldsymbol{q}_n^g, g(\boldsymbol{I}) \rangle. \tag{8}$$

As such, the desired equivariance property of both feature representations and instance query embeddings to input transformations are formulated in a unified training objective. For Eq. 8, the queries $\{\boldsymbol{q}_n^g\}_{n=1}^N$, created for transformed image $g(I)$, are first applied for the transformed representation $g(\boldsymbol{I})$ of $I$; then the training target is to minimize the difference between the retrieved instance masks $\{\hat{W}_n^g \in [0, 1]^{H \times W}\}_{n=1}^N$ and the transformed groundtruth masks $\{g(M_{\sigma(n)})\}_{n=1}^N$. Therefore, $\mathcal{L}_{equi}$ can be easily implemented as any existing mask prediction loss, and effortless incorporated into our training framework. For the sake of simplicity, we set $\mathcal{L}_{equi}$ as the same form of $\mathcal{L}_{mask}$ in Eq. 3. In our experiments (*cf*. §4.3), we will demonstrate that our equivariance constraint can bring significantly larger performance gains than the conventional, transformation-based data augmentation technique.

### 3.3 Implementation Detail

**External Memory and Sampling.** For large-scale query-instance matching (*cf*. Eq. 5), we build an external memory. Due to the limited capacity of our GPU, we afford to maintain a queue of 100K pixel samples in the memory. We adopt sparse, instance-balanced sampling, *i.e.*, randomly selecting 50 (if possible) pixels from each instance. Note that the samples stored in the memory are only used for the computation of the inter-scene instance discrimination loss $\mathcal{L}_{inter\_mask}$ in Eq. 5. The external memory is directly discarded after training, causing no extra computation budget during inference.

**Training Objective.** Ordinarily, our training framework is applicable for existing query-based models, and complementary to their training objectives. In our experiments, we approach our algorithm on four representative query-based methods [34,39–41] by adding our inter-scene instance discrimination loss $\mathcal{L}_{inter\_mask}$ (*cf*. Eq. 5) and equivariance loss $\mathcal{L}_{equi}$ (*cf*. Eq. 8) into their training targets. As mentioned before, we implement $\mathcal{L}_{inter\_mask}$ (*cf*. Eq. 5) as the focal loss [66], whose hyper-parameters are set to $\alpha = 0.1$ and $\gamma = 2.5$. As for $\mathcal{L}_{equi}$, it is directly set as the same form of the loss used for mask prediction in the base segmenter, *i.e.*, dice loss [67] for [34, 39, 41], and a linear combination of the focal loss and dice loss for [40]. For far comparison, the set of transformations $G$, which is used for the computation of $\mathcal{L}_{equi}$, is {horizontal flipping, random cropping between 0.6 and 1.0}. This is consistent with the standard data augmentation setup adopted by existing instance segmentation networks. To balance the impacts of our two new training targets, *i.e.*, $\mathcal{L}_{inter\_mask}$ and $\mathcal{L}_{equi}$, we multiply $\mathcal{L}_{equi}$ by a coefficient $\lambda$, which is empirically set as 3 (see related experiments in Table 4e & §4.3).

# 4 Experiments

## 4.1 Experimental Setup

**Datasets.** We conduct our main experiments on the gold-standard benchmark dataset, *i.e.*, COCO[47], in this field. There are 80 target classes for instance segmentation. As normal, we use `train2017` split (115k images) for training and `val2017` (5k images) for validation used in our ablation study. The main results are reported on `test-dev` (20k images). For thorough evaluation, we perform additional experiments on LVISv1[48] on top of SOLOv2[39]. LVIS is a large vocabulary instance segmentation dataset collected by re-annotating COCO images with 1,203 categories. Thus it is considerably more challenging; it contains a total of 100k images of a `train` set under a significant long-tailed distribution, and relatively balanced `val` (20k images) and `test` sets (20k images).

**Base Instance Segmenters, Backbones, and Competitors.** To demonstrate the broad applicability and wide benefit of our algorithm, we approach our method on four widely recognized query-based instance segmentation models, *i.e.*, CondInst[34], SOLOv2[39], SOTR[41], and Mask2Former[40], with diverse backbone networks, *i.e.*, ResNet-50/-101[45] and Swin-Small/-Base/-Large[46]. For fair comparison, all these models and results are based on our reproduction, following the default hyper-parameter and augmentation recipes in AdelaiDet[76] and MMDetection[77]. In addition to focusing on the comparison with these four base segmentation models, we include a group of famous segmenters [6, 11, 12, 15, 18, 43, 44] for comprehensive evaluation. ***In our supplementary material, we report more experiments on two instance segmentation models, i.e., SparseInst[44] and SOLQ[43], as well as performance of Mask2Former on panoptic segmentation.***

**Training.** Our implementation is based on AdelaiDet[76] and MMDetection[77], following default training configurations for both COCO and LVISv1 datasets. In particular, all the backbones are initialized using corresponding weights pre-trained on ImageNet-1K[78], while remaining layers are randomly initialized. We train models using SGD with initial learning rate of 0.01 for ResNet[45] backboned models and Adam with initial learning rate of $1e^{-5}$ for Swin[46] backboned models. The learning rate is scheduled following the polynomial annealing policy with batch size 16. We use multi-scale training [12, 16, 34, 44]: for ResNet backboned models, the short side of input is randomly chosen within [400, 1200] and long side is set to 1333; for Swin backboned models, we opt random scaling with a factor in [0.1, 2.0] followed by a fixed size to $1024 \times 1024$. ResNet backboned models are trained for 12 epochs, while Swin backboned models are trained for 50 epochs. Note that, compared with the standard, scene-wise training strategy, our algorithm only slows the training speed slightly ($\sim$5%; see Table 3 & §4.3).

**Testing.** All our experimental results are reported with $1333 \times 800$ input resolution. For the sake of fairness, *we do not apply any test-time data augmentation*. Note that, during model deployment, our training algorithm does not bring any change to network architecture or additional computation cost.

**Evaluation Metric.** The evaluation metric is the standard segmentation mask *AP*.

**Reproducibility.** Our algorithm is implemented in PyTorch. For all our experiments, the training and testing are conducted on eight NVIDIA Tesla A100 GPUs with a 80GB memory per-card. To guarantee the reproducibility of our algorithm, our full implementations are made publicly available at https://github.com/JamesLiang819/Instance_Unique_Querying.

## 4.2 Comparison to State-of-the-Arts

**Quantitative Results on COCO `test-dev`.** Table 1 reports comparison results with our four base models, *i.e.*, CondInst[34], SOLOv2[39], SOTR[41], and Mask2Former[40], as well as several representative instance segmentation methods [6, 11, 12, 15, 18, 43, 44, 79] on COCO[47] `test-dev`. Without bells and whistles, our training framework provides remarkable performance improvements over the four query-based segmenters [34, 39–41] with different backbones. In particular, for CondInst[34] and SOLOv2[39] — the two famous query-based approaches that adopt the CNN architecture and dynamic neural network, with ResNet-50[45] backbone, our algorithm brings **3.1** and **3.2** *AP* gains, respectively. Similar improvements, *i.e.*, **2.8** and **2.9** in terms of *AP*, are obtained with ResNet-101. As for SOTR[41] — a very recent method that adopts a hybrid structure of Transformer[80] structure and dynamic convolution for query creation, our algorithm also greatly promotes its performance, *e.g.*, 39.6→**42.2** with ResNet-50 and 40.2→**42.6** with ResNet-101, in terms of *AP*. Moreover, when directly applying our algorithm to the newest query-based model — Mask2Former[40], we observe powerful

Table 1: Quantitative results on COCO [47] `test-dev`. See §4.2 for details.

| Method | Backbone | #Epoch | $AP$ | $AP_{50}$ | $AP_{75}$ | $AP_S$ | $AP_M$ | $AP_L$ |
|---|---|---|---|---|---|---|---|---|
| Mask R-CNN[ICCV17] [6] | ResNet-101 | 12 | 36.1 | 57.5 | 38.6 | 18.8 | 39.7 | 49.5 |
| Cascade Mask R-CNN[PAMI19] [11] | ResNet-101 | 12 | 37.3 | 58.2 | 40.1 | 19.7 | 40.6 | 51.5 |
| HTC[CVPR19] [12] | ResNet-101 | 20 | 39.6 | 61.0 | 42.8 | 21.3 | 42.9 | 55.0 |
| Point Rend[CVPR20] [15] | ResNet-50 | 12 | 36.3 | 56.9 | 38.7 | 19.8 | 39.4 | 48.5 |
| QueryInst[ICCV21] [18] | ResNet-101 | 36 | 41.0 | 63.3 | 44.5 | 21.7 | 44.4 | 60.7 |
| K-Net[NeurIPS21] [79] | ResNet-101 | 36 | 40.1 | 62.8 | 43.1 | 18.7 | 42.7 | 58.8 |
| SOLQ[NeurIPS21] [43] | Swin-L | 50 | 46.7 | 72.7 | 50.6 | 29.2 | 50.1 | 60.9 |
| SparseInst[CVPR22] [44] | ResNet-50 | 36 | 37.9 | 59.2 | 40.2 | 15.7 | 39.4 | 56.9 |
| CondInst[ECCV20] [34] | ResNet-50 | | 35.5 | 55.8 | 37.7 | 16.8 | 39.2 | 50.6 |
| +Ours | | 12 | 38.6↑3.1 | 61.1↑5.3 | 41.2↑3.5 | 19.7↑2.9 | 41.1↑1.9 | 54.7↑4.1 |
| CondInst[ECCV20] [34] | ResNet-101 | | 37.1 | 58.6 | 39.3 | 18.2 | 40.3 | 52.9 |
| +Ours | | | 39.9↑2.8 | 62.7↑4.1 | 42.4↑3.1 | 20.8↑2.6 | 42.3↑2.0 | 55.7↑2.8 |
| SOLOv2[NeurIPS20] [39] | ResNet-50 | | 35.1 | 55.5 | 37.0 | 13.9 | 38.4 | 53.7 |
| +Ours | | 12 | 38.3↑3.2 | 59.6↑4.1 | 40.6↑3.6 | 17.8↑3.9 | 41.8↑3.4 | 56.2↑2.5 |
| SOLOv2[NeurIPS20] [39] | ResNet-101 | | 36.7 | 57.5 | 39.3 | 16.2 | 40.2 | 54.2 |
| +Ours | | | 39.6↑2.9 | 60.6↑3.1 | 43.1↑3.8 | 20.0↑3.8 | 43.3↑3.1 | 56.9↑2.7 |
| SOTR[ICCV21] [41] | ResNet-50 | | 39.6 | 60.7 | 42.6 | 10.3 | 58.7 | 72.1 |
| +Ours | | 24 | 42.2↑2.6 | 61.9↑3.1 | 43.9↑2.3 | 11.0↑0.7 | 60.5↑2.8 | 73.5↑2.4 |
| SOTR[ICCV21] [41] | ResNet-101 | | 40.2 | 61.2 | 43.4 | 10.2 | 59.0 | 73.1 |
| +Ours | | | 42.6↑2.4 | 64.1↑2.9 | 45.8↑2.4 | 11.2↑1.0 | 61.2↑2.2 | 75.3↑2.2 |
| Mask2Former[CVPR22] [40] | Swin-S | | 44.5 | 68.2 | 47.7 | 24.3 | 49.0 | 66.6 |
| +Ours | | 50 | 46.7↑2.2 | 70.7↑2.5 | 50.1↑2.4 | 27.0↑2.7 | 50.9↑1.9 | 68.8↑2.2 |
| Mask2Former[CVPR22] [40] | Swin-B | | 44.9 | 68.5 | 48.5 | 24.6 | 48.7 | 67.4 |
| +Ours | | | 47.3↑2.4 | 71.3↑2.8 | 51.0↑2.5 | 28.3↑2.7 | 50.9↑2.2 | 69.4↑2.0 |
| Mask2Former[CVPR22] [40] | Swin-L | 100 | 50.2 | 74.8 | 54.7 | 29.2 | 53.8 | 71.1 |
| +Ours | | | 51.8↑1.6 | 76.0↑1.2 | 56.8↑2.1 | 29.9↑0.7 | 55.1↑1.3 | 73.3↑2.2 |

Table 2: Quantitative results on LVISv1 [48] `val` over SOLOv2 [39]. See §4.2 for details.

| Method | Backbone | #Epoch | $AP$ | $AP_{50}$ | $AP_{75}$ | $AP_r$ | $AP_c$ | $AP_f$ |
|---|---|---|---|---|---|---|---|---|
| Mask R-CNN[ICCV17] [6] | ResNet-50 | 12 | 21.7 | 34.3 | 23.0 | 9.6 | 21.0 | 27.8 |
| SOLOv2[NeurIPS20] [39] | ResNet-50 | 36 | 21.4 | 34.0 | 22.8 | 9.5 | 20.9 | 27.6 |
| +Ours | | | 24.1↑2.7 | 37.4↑3.4 | 25.5↑2.7 | 13.5↑4.0 | 22.8↑1.9 | 29.7↑2.1 |

performance gains across different Swin [46] backbones. For instance, with Swin-B, our algorithm surpasses the vanilla Mask2Former at different IoU thresholds, *i.e.*, **71.3** *vs* 68.5 $AP_{50}$ and **51.0** *vs* 48.5 $AP_{75}$. Notably, with Swin-L as the Mask2Former's backbone, our approach surpasses all the other competitors in Table 1 and sets a new state-of-the-art record of **51.8** in $AP$ for instance segmentation on COCO. Furthermore, in spite of different base segmentation network architectures and backbones, our training algorithm consistently improves the performance for object instances of different sizes.

**Quantitative Results on LVISv1 [48] `val`.** Table 2 lists the overall performance on LVISv1 `val`, as well as $AP$ scores on the *rare* (1∼10 instances), *common* (11∼100 instances), and *frequent* (> 100 instances) subsets. It can be observed that, our training algorithm pushes a powerful gain by **2.7** on $AP$, and consistently improves the performance of the base model, SOLOv2, on $AP_r$, $AP_c$, and $AP_f$ by **4.0**, **1.9** and **2.1**, respectively.

The above experimental results are particularly impressive, considering the fact that the performance gain is purely brought by a new training strategy, without any network architectural modification. Overall, our extensive experiments manifest the effectiveness and wide benefit of our algorithm.

**Qualitative Result.** Fig. 3 provides qualitative comparisons of our algorithm against CondInst [34] (left) and Mask2Former [40] (right) on several challenging examples of COCO `val2017`. As can be seen, with the help of our training algorithm, CondInst and Mask2Former can better distinguish between huddled and similar instances, hence generating higher-quality segmentation maps.

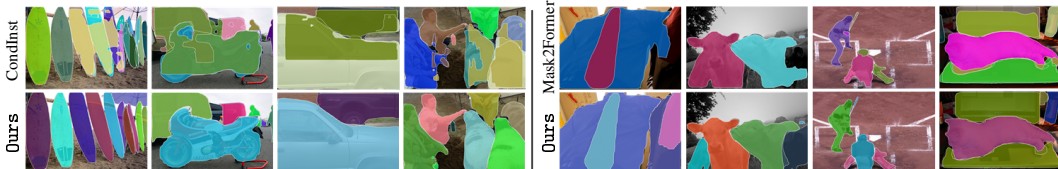

Figure 3: **Qualitative comparison** results on COCO `val2017`. See §4.2 for details.

Table 3: Analysis of essential components on COCO [47] `val2017`. See §4.3 for details.

| Inter-scene $\mathcal{L}_{inter\_mask}$ (Eq. 5) | Equivariance $\mathcal{L}_{equi}$ (Eq. 8) | $AP$ | $AP_{50}$ | $AP_{75}$ | $AP_S$ | $AP_M$ | $AP_L$ | Training speed (hour/epoch) |
|:---:|:---:|:---:|:---:|:---:|:---:|:---:|:---:|:---:|
| | | 35.5 | 55.9 | 37.4 | 16.9 | 38.9 | 50.2 | 1.51 |
| ✓ | | 37.6 | 58.4 | 40.3 | 18.2 | 40.8 | 54.0 | 1.57 |
| | ✓ | 36.5 | 58.0 | 39.0 | 17.1 | 39.5 | 51.6 | 1.52 |
| ✓ | ✓ | $38.1_{\uparrow 2.6}$ | $60.2_{\uparrow 4.3}$ | $40.6_{\uparrow 3.2}$ | $18.4_{\uparrow 1.5}$ | $40.9_{\uparrow 2.0}$ | $54.3_{\uparrow 4.1}$ | 1.59 |

## 4.3 Diagnostic Experiment

For thorough examination, we conduct a series of ablative studies on COCO [47] `val2017`. We adopt CondInst [34] as our base segmentation network with ResNet-50 [45] backbone. To perform extensive experiments, we train models for 12 epochs while keeping other hyper-parameters unchanged.

**Key Component Analysis.** First, we study the efficacy of the core components of our algorithm, *i.e.*, inter-scene instance discrimination loss $\mathcal{L}_{inter\_mask}$ (*cf.* Eq. 5) and equivariance loss $\mathcal{L}_{equi}$ (*cf.* Eq. 8). In Table 3, 1*st* row gives the performance of our base segmenter CondInst. For 2*nd* and 3*rd* rows, the scores are obtained by adopting $\mathcal{L}_{inter\_mask}$ and $\mathcal{L}_{equi}$ individually. And 4*th* row reports the scores of our full algorithm. **i)** 1*st vs* 2*nd* row: $\mathcal{L}_{inter\_mask}$ leads to notable performance improvements against the baseline (*e.g.*, 35.5→**37.6** *AP*). This verifies our first core hypothesis, *i.e.*, learning large-scale instance separation benefits discriminative query embedding learning. **ii)** 1*st vs* 3*rd* row: $\mathcal{L}_{equi}$ greatly boots the performance of the baseline (*e.g.*, 35.5→**36.5** *AP*), evidencing our second core hypothesis, *i.e.*, exploiting the transformation equivariance nature of the task facilitates robust query-instance matching. **iii)** 1*st vs* 2*nd vs* 3*rd vs* 4*th* row: combing all contributions together results in the largest gain over the baseline (*e.g.*, 35.5→**38.1** *AP*). This suggests that $\mathcal{L}_{inter\_mask}$ and $\mathcal{L}_{equi}$ are able to work in a collaborative manner, and confirms the effectiveness of our overall algorithmic design.

**Training Speed.** We also report training speed in Table 3; related statistics are gathered on eight A100 GPUs with a batch size of 16. We can find that, compared with the current scene-wise training regime, our cross-scene training algorithm only brings slight delay (∼5%). More specifically, the computation of our inter-scene instance discrimination loss $\mathcal{L}_{inter\_mask}$ takes relatively more time, due to the use of an external memory for large-scale query-instance matching.

Next we study some core designs for inter-scene instance discrimination loss $\mathcal{L}_{inter\_mask}$ (*cf.* Eq. 5). The results in Tables 4a, 4b, and 4c are reported without considering equivariance loss $\mathcal{L}_{equi}$ (*cf.* Eq. 8).

**Loss Design for Inter-Scene Instance Uniqueness Learning.** In Table 4a, we investigate four different implementation forms of our inter-scene instance discrimination loss $\mathcal{L}_{inter\_mask}$, namely $\ell_1$ loss, $\ell_2$ loss, cross-entropy loss, and focal loss [66]. We can find that focal loss is more favored. In our case, most instance (pixel) samples from other scenes can be easily recognized as negative, causing a huge imbalance between hard and easy negative samples. Therefore, training under such an environment results in $\ell_1$ loss, $\ell_2$ loss, and cross-entropy loss focusing more on easy samples. In contrast, focal loss drives the training more towards the sparse set of hard negative samples, thus preventing the gradient of $\mathcal{L}_{inter\_mask}$ from being dominated by the massive, easy negative samples.

**Sampling Strategy.** We further study the impact of the three sampling strategies (*cf.* §3.2) for the computation of $\mathcal{L}_{inter\_mask}$, namely dense sampling (storing all the pixels of each image into the memory), sparse sampling (randomly sampling a small set, *e.g.*, 0.5k or 1.0k, of pixels from each image), and instance-balanced sampling (randomly sampling a small fixed-size set, *e.g.*, 10 or 50, of pixels from each instance). Table 4b proves that, sparse sampling works overall better than dense sampling, as it improves the diversity of the stored samples. However, sparse sampling leads to inferior

Table 4: A set of ablative studies on COCO [47] `val2017`. The adopted algorithm designs and hyper-parameter settings are marked in red. See §4.3 for details.

| $\mathcal{L}_{inter\_mask}$ | $AP$ | $AP_S$ | $AP_M$ | $AP_L$ |
|---|---|---|---|---|
| - | 35.5 | 16.9 | 38.9 | 50.2 |
| $\ell_1$ loss | 36.0 | 17.1 | 39.2 | 51.7 |
| $\ell_2$ loss | 35.9 | 17.6 | 39.1 | 51.6 |
| cross-entropy | 37.3 | 18.1 | 40.6 | 53.3 |
| focal loss | 37.6 | 18.6 | 40.6 | 54.0 |

(a) inter-scene instance discrimination loss $\mathcal{L}_{inter\_mask}$ (*cf.* Eq. 5)

| sampling | $AP$ | $AP_S$ | $AP_M$ | $AP_L$ |
|---|---|---|---|---|
| dense | 37.1 | 18.0 | 40.4 | 53.0 |
| sparse (0.5k pixel/img.) | 37.2 | 17.8 | 40.5 | 53.2 |
| sparse (1.0k pixel/img.) | 37.3 | 17.9 | 40.5 | 53.4 |
| instance-balanced (10 pixel/ins.) | 37.4 | 18.7 | 40.5 | 53.6 |
| instance-balanced (50 pixel/ins.) | 37.6 | 18.6 | 40.6 | 54.0 |

(b) sampling strategy

| capacity | $AP$ | $AP_S$ | $AP_M$ | $AP_L$ |
|---|---|---|---|---|
| - | 35.5 | 16.9 | 38.9 | 50.2 |
| mini-batch (w/o memory) | 36.9 | 17.8 | 40.1 | 53.1 |
| 10k pixels | 37.0 | 17.9 | 40.3 | 53.2 |
| 50k pixels | 37.3 | 18.1 | 40.4 | 53.6 |
| 70k pixels | 37.5 | 18.5 | 40.4 | 53.7 |
| 100k pixels | 37.6 | 18.6 | 40.6 | 54.0 |

(c) memory capacity

| transformation equivariance | $AP$ | $AP_S$ | $AP_M$ | $AP_L$ |
|---|---|---|---|---|
| *w/o* Aug. | 35.3 | 16.2 | 39.2 | 49.9 |
| *w/* Aug. ($\{\langle q_n^g, I^g\rangle\}_n \approx \{g(M_{\sigma(n)})\}_n$) | 35.5 | 16.9 | 38.9 | 50.2 |
| feature *only* ($I^g \approx g(I)$) | 36.3 | 16.5 | 39.5 | 51.4 |
| $\mathcal{L}_{equi}$ ($\{\langle q_n^g, g(I)\rangle\}_n \approx \{g(M_{\sigma(n)})\}_n$) | 36.5 | 17.1 | 39.5 | 51.6 |

(d) transformation equivariance modeling

| coefficient | $AP$ | $AP_S$ | $AP_M$ | $AP_L$ |
|---|---|---|---|---|
| 1 | 37.0 | 17.9 | 40.4 | 52.1 |
| 3 | 37.6 | 18.6 | 40.6 | 54.0 |
| 5 | 37.5 | 18.3 | 40.7 | 53.6 |
| 10 | 37.3 | 18.1 | 40.8 | 53.3 |

(e) coefficient between $\mathcal{L}_{inter\_mask}$ and $\mathcal{L}_{equi}$

performance on small instances, *e.g.*, 18.0 *vs* 17.8 $AP_S$ for 'sparse (0.5k pixel/img.)', as smaller instances are less likely sampled for loss computation. Fortunately, instance-balanced sampling inherits sparse sampling's desired merits but without its defects, hence yielding more impressive results.

**Memory Capacity.** Table 4c shows the influence of the capacity of the external memory. 1*st* row gives the results of the base segmenter. 2*nd* row lists the scores obtained by computing $\mathcal{L}_{inter\_mask}$ within each mini-batch, which are already much better, *e.g.*, 35.5 *vs* **36.9** $AP$. 3*rd* - 5*th* rows demonstrate that, **i)** with the aid of an external memory, the performance can be further boosted; and **ii)** enlarging memory is always helpful. These results verify our key idea of learning large-scale query-instance matching. Note that the optimal configuration, *i.e.*, 100k pixels, does not yet reach the point of performance saturation, but rather the upper limit of our hardware's computational budget.

**Transformation Equivariant Learning.** We next study our equivariance loss design $\mathcal{L}_{equi}$ (*cf.* Eq. 8). The results in Table 4d are reported without considering the inter-scene instance discrimination loss $\mathcal{L}_{inter\_mask}$ (*cf.* Eq. 5). The first two rows respectively present the performance of the base segmenter *w/o* and *w/* transformation-based data augmentation. As we mentioned before, training with current data augmentation technique can be viewed as learning equivariant query embeddings (*cf.* Eq. 7). In 3*rd* row we show the performance by only encouraging equivariant image representation learning (*cf.* Eq. 6). Comparing these three baselines with our final equivariance loss $\mathcal{L}_{equi}$ that simultaneously addresses equivariant image representation and query embedding learning, we can find: **i)** exploring the equivariance nature of the task is indeed essential; and **ii)** our equivariance loss $\mathcal{L}_{equi}$ is much more effective than current transformation-based data augmentation technique (*e.g.*, **36.5** *vs* 35.5 $AP$).

**Loss Term Coefficient.** For completeness, the results with different coefficients, which control the balance between our two training objectives, *i.e.*, $\mathcal{L}_{inter\_mask}$ and $\mathcal{L}_{equi}$, are reported in Table 4e.

## 5   Conclusion and Discussion

Aiming at sharpening the instance discrimination ability of query-based segmenters, we devise a novel framework that formulates two crucial properties of the query-instance relationship, *i.e.*, *dataset-level uniqueness* and *transformation equivariance*, as network training targets. This is achieved by compelling segmenters to i) build exclusive instance-query matching throughout the entire training dataset, and ii) learn equivariant instance-query matching with respect to geometric transformations. Extensive empirical analysis demonstrates that our method is flexible yet powerful, allowing it to benefit the existing and growing body of query-based instance segmentation methods. We argue deductively that the proposed training framework has the potential to be applied to broader range of dense prediction tasks, *i.e.*, query-based detection and panoptic segmentation. These questions remain open for our future endeavor.

**Acknowledgement.** This work was partially supported by ARC DECRA DE220101390.

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
