# Learning Equivariant Segmentation with Instance-Unique Querying
## *Supplementary Material*

**Wenguan Wang**[*]
ReLER, AAII, University of Technology Sydney

**James Liang**[*]
Rochester Institute of Technology

**Dongfang Liu**[†]
Rochester Institute of Technology

This document provides additional experimental results, more details of our approach and discussion, organized as follows:

- §S1: More quantitative results
- §S2: More qualitative results
- §S3: Pseudo code of our algorithm
- §S4: Further discussion

## S1 More Quantitative Results

Table S1: **More Quantitative Results** on COCO [1] `test-dev`. See §S1 for details.

| Method | Backbone | #Epoch | $AP$ | $AP_{50}$ | $AP_{75}$ | $AP_S$ | $AP_M$ | $AP_L$ |
|---|---|---|---|---|---|---|---|---|
| SparseInst[CVPR22] [2] | ResNet-50 | 36 | 34.7 | 55.3 | 36.6 | 14.3 | 36.2 | 50.7 |
| +Ours | | | **36.7**↑2.0 | **57.1**↑1.8 | **38.8**↑2.2 | **16.1**↑1.8 | **38.9**↑2.7 | **56.7**↑6.0 |
| SparseInst[CVPR22] [2] | ResNet-50-DCN | | 35.4 | 56.4 | 37.2 | 15.7 | 36.7 | 53.9 |
| +Ours | | | **37.7**↑2.3 | **58.7**↑2.3 | **39.7**↑2.5 | **17.9**↑2.2 | **39.5**↑2.8 | **58.3**↑4.4 |
| SOLQ[NeurIPS21] [3] | ResNet-50 | 50 | 39.7 | 63.3 | 42.6 | 21.5 | 42.5 | 53.1 |
| +Ours | | | **41.9**↑2.2 | **65.3**↑2.0 | **44.9**↑2.3 | **23.0**↑1.5 | **44.8**↑2.3 | **56.3**↑3.2 |
| SOLQ[NeurIPS21] [3] | Swin-L | | 46.7 | 72.7 | 50.6 | 29.2 | 50.1 | 60.9 |
| +Ours | | | **48.7**↑2.0 | **74.4**↑1.7 | **52.8**↑2.2 | **30.8**↑1.6 | **52.1**↑2.0 | **63.4**↑2.5 |

**More Base Instance Segmenters.** To further demonstrate the power of our methodology, we apply our training algorithm to two additional, concurrent query-based instance segmentation models (*i.e.*, SparseInst [2] and SOLQ [3]), with their default hyperparameter settings. The results on the COCO `test-dev` are reported in Table S1. SparseInst [2] is a fast segmenter that learns a sparse set of instance-aware queries and predicts instances in a one-to-one style without non-maximum suppression. As seen, with the help of our algorithm, the performance is significantly boosted to **36.7** and **37.7** *AP* with ResNet-50 [4] and ResNet-50-DCN [5] backbones, which are **2.0** and **2.3** higher than the baseline, respectively. SOLQ [3] is a very recent segmenter that learns a unified query representation to directly predict the class, location, and mask of the instance. Similarly, our algorithm greatly promotes the performance by **2.2** *AP* correspondingly with ResNet-50. On the top of SOLQ, we also test our algorithm on the strong backbone — Swin [6] backbone. Without bells and whistles, our algorithm also yields a consistent **2.0** improvement in *AP*, a strong indication that the proposed method is also compatible with transformer-based backbone network architecture.

---

[*]authors contributed equally

[†]corresponding author

36th Conference on Neural Information Processing Systems (NeurIPS 2022).

Table S2: **Quantitative results** on COCO Panoptic [1] `val` over Mask2Former [7]. See §S1 for details.

| Method | Backbone | #Epoch | $PQ$ | $PQ^{th}$ | $PQ^{st}$ |
|---|---|---|---|---|---|
| Mask2Former | Swin-B | 50 | 56.1 | 62.5 | 46.7 |
| +Ours | | | **57.3**$_{\uparrow 1.2}$ | **64.1**$_{\uparrow 1.6}$ | **47.4**$_{\uparrow 0.7}$ |

Table S3: **Ablative study** of additional data augmentation on COCO [1] `val2017` over CondInst [9] with ResNet-50. See §S1 for details.

| Method | Data Augmentation | #Epoch | $AP$ | $AP_S$ | $AP_M$ | $AP_L$ |
|---|---|---|---|---|---|---|
| CondInst | - | 12 | 35.3 | 16.2 | 39.2 | 49.9 |
| CondInst | `flip+crop` | 12 | 35.5 | 16.9 | 38.9 | 50.2 |
| +Ours | | | 36.5 | 17.1 | 39.5 | 51.6 |
| CondInst | `flip+crop+rotation` | 12 | 35.6 | 16.9 | 39.2 | 50.5 |
| +Ours | | | 36.9 | 17.1 | 39.6 | 52.4 |
| CondInst | `flip+crop+scaling` | 12 | 35.9 | 17.0 | 39.6 | 51.4 |
| +Ours | | | 37.3 | 17.4 | 40.1 | 53.3 |
| CondInst | `flip+crop+rotation+scaling` | 12 | 35.9 | 17.0 | 39.7 | 51.7 |
| +Ours | | | 37.6 | 17.5 | 40.3 | 54.5 |

The above observations suggest that our algorithm is generic for query-based models regardless of their specific segmentation architectures and backbones, and consistently provides performance enhancements.

**Additional Panoptic Segmentation Task.** To reveal the power of our idea and further demonstrate the high versatility of our algorithm, we run experiments on the panoptic segmentation task. In particular, panoptic segmentation includes the extraction of both *things* (*i.e.*, instance segmentation) and *stuffs* (*i.e.*, semantic segmentation). To accommodate the algorithmic setting, the only modification that needs to be made is for the stuff classes. Since the segmentation of the stuff classes is instance-agnostic, the training objectives for cross-scene querying (*cf.* Eq. 5) of the stuff classes should be the ground-truth stuff masks, instead of all-zero matrices used for thing classes/instance segmentation. In Table S2, we report the results on MS COCO Panoptic [1], on top of Mask2Former [8]. Empirically, our margins over the baseline are significant, *i.e.*, **1.2**, **1.6**, and **0.7** on $PQ$, $PQ^{th}$, and $PQ^{st}$, respectively.

**More Ablation Studies.** To further test the efficacy of our transformation equivariant learning strategy, we run experiments on an enlarged transformation family, namely {horizontal flipping, random cropping between 0.6 and 1.0, random rotation between $0°$ to $60°$, random scaling between 0.5 to 2.0}. Note that rotation and scale transformations are not included in our main experiments, as they are not the default data augmentation operations in MMDetection. Table S3 provides a point-to-point comparison, where the first three baselines correspond to the first, second, and last rows of Table 3d, respectively, while the last six baselines are newly added. All the experiments are conducted with the equivariance loss $\mathcal{L}_{equi}$ (*cf.* Eq. 8) *only*. As seen, our transformation equivariant learning strategy further promotes the performance with more transformation operations, and is much more effective than the classic, transformation-based data augmentation strategy, *i.e.*, simply treating transformed images as new training samples. Moreover, we also report the training speed and GPU memory cost for different sizes of the external memory capacity in Table S4. As seen, the optimal configuration of the memory capacity, *i.e.*, 100k pixels, only causes marginal training speed delay ($\sim 5\%$) as a trade-off but achieves a significant gain of **2.1** on mask $AP$.

## S2 More Qualitative Results

In Fig. S1 and Fig. S2, we show more qualitative results on COCO `val2017` over CondInst [9] and Mask2Former [8], respectively. As has been observed, our approach consistently produces more accurate predictions than the baseline. For instance, in the bus example (Fig. S1 row 4), the baseline model wrongly divides the center bus into two instances; in the giraffe example (Fig. S2 row 5), the baseline model fails to separate the two giraffes. However, our algorithm works well in

Table S4: **Ablative study** of memory capacity. See §S1 for details.

| capacity | $AP$ | $AP_S$ | $AP_M$ | $AP_L$ | Training speed (minutes/epoch) | GPU memory cost (GB) |
|---|---|---|---|---|---|---|
| - | 35.5 | 16.9 | 38.9 | 50.2 | 90 | 19.68 |
| mini-batch (*w/o* memory) | 36.9 | 17.8 | 40.1 | 53.1 | 106 | 19.68 |
| 10k pixels | 37.0 | 17.9 | 40.3 | 53.2 | 91 | 20.24 |
| 50k pixels | 37.3 | 18.1 | 40.4 | 53.6 | 93 | 22.37 |
| 70k pixels | 37.5 | 18.5 | 40.4 | 53.7 | 93 | 23.72 |
| 100k pixels | 37.6 | 18.6 | 40.6 | 54.0 | 94 | 25.37 |

these challenging scenarios, confirming its ability of learning more discriminative instance query embeddings.

**Failure Case Analysis.** Though greatly promoting the instance segmentation performance, our algorithm also struggles with some extremely challenging scenarios. Fig. S3 summarizes the most representative failure cases and concludes their characteristic patterns that may cause the inferior results. Specifically, we observe that the failure cases are mostly found in the following cases: i) highly similar and occluded object instances; ii) object instances with complex topologies; iii) small and blurry instances; iv) highly deformed object instances; and v) transparent object instances. We argue that one critical reason for the erroneous prediction is that the object query would be confused under these conditions. In particular, for highly occluded objects in query-based models, objects with the same labels and high occlusion may share very similar queries. Though our algorithm faces difficulties in these scenarios, it has led to a significant improvement compared with the baseline model. In the meanwhile, the patterns of these failure cases also shed light on the possible direction of our future efforts.

# S3 Pseudo Code

The pseudo-code of our algorithm is given in Alg. 1.

# S4 Discussion

**Broader Impact.** This work proposes a new training scheme for discriminating between instances across scenes and against geometric transformations, so as to achieve effective instance separation. Our algorithm has demonstrated its effectiveness over a variety of modern query-based instance segmenters. On the positive side, the research may have a wide range of real-world applications, including self-driving cars, robot navigation, and medical imaging. On the negative side, any inaccurate prediction in real-world applications (*e.g.*, autonomous driving tasks and medical imaging analysis) raises concerns about human safety. To avoid this potential negative societal impact, we suggest proposing an extremely strict security protocol in case of dysfunction of our method in real-world applications.

**Limitation Analysis.** One limitation of our algorithm arises from the restriction of equivariant transformations that have to be elements from a group of linear transformation operators. Therefore, we can only apply ordinary linear transformation (*i.e.*, flipping and cropping) while arbitrary photometric transformation (*i.e.*, color jittering and blur) is not suitable for our algorithm. We will put more effort into finding appropriate candidates for transformation operators and evaluating their performance with them for instance segmentation. In addition, we aim to explore a more effective equivariance training strategy by utilizing the most recent developments in equivariance representation learning.

**Algorithm 1** Pseudo-code of our method in a PyTorch-like style.

```
# img: input image in shape (batch_size, c, h, w)
# query: query from decoder in shape (num_queries, batch_size, c)
# external_feats: external features from the upstream external memory bank,
    each is a 4D-tensor
# lambda: hyper-parameter for the equivariance loss

# Inter_mask prediction loss (Eq.5) #
def Inter_mask_Loss(query, external_feats):

    # Feature embedding
    mask_embed = mask_embed(query)

    # Loss calculation
    mask_pred = torch.einsum('bqc,nchw->bnqhw',
    mask_embed, external_feats.clone().detach())
    inter_mask_loss = Focal_Loss(mask_pred, torch.zeros_like(mask_pred))

    return inter_mask_loss

# Equivariance loss (Eq.8) #
def Equivariance_Loss(img, lambda = 3)

    # Get feats from images
    feats = encoder(img)

    # Apply randomly from a list of transformations
    transforms = torch.nn.ModuleList([torchvision.transforms.
        RandomHorizontalFlip, torchvision.transforms.RandomCrop])
    transform = torchvision.transforms.RandomChoice(transforms)

    # Get transformed feats and query
    transformed_feats = transform(feats)
    transformed_query = decoder(encoder(transform(img)))

    # Get ground truth from transformed image
    transformed_gt_mask = transform(GetGroundtruth(img))

    # Get mask prediction from the model
    transformed_mask_embed = mask_embed(transformed_query)
    transformed_mask_pred = torch.einsum('bqc,bchw->bqhw',
    mask_embed, transformed_feats)

    # Loss calculation
    equivariance_loss = criterion(transformed_mask_pred,transformed_gt_mask)

    return lambda*equivariance_loss
```

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

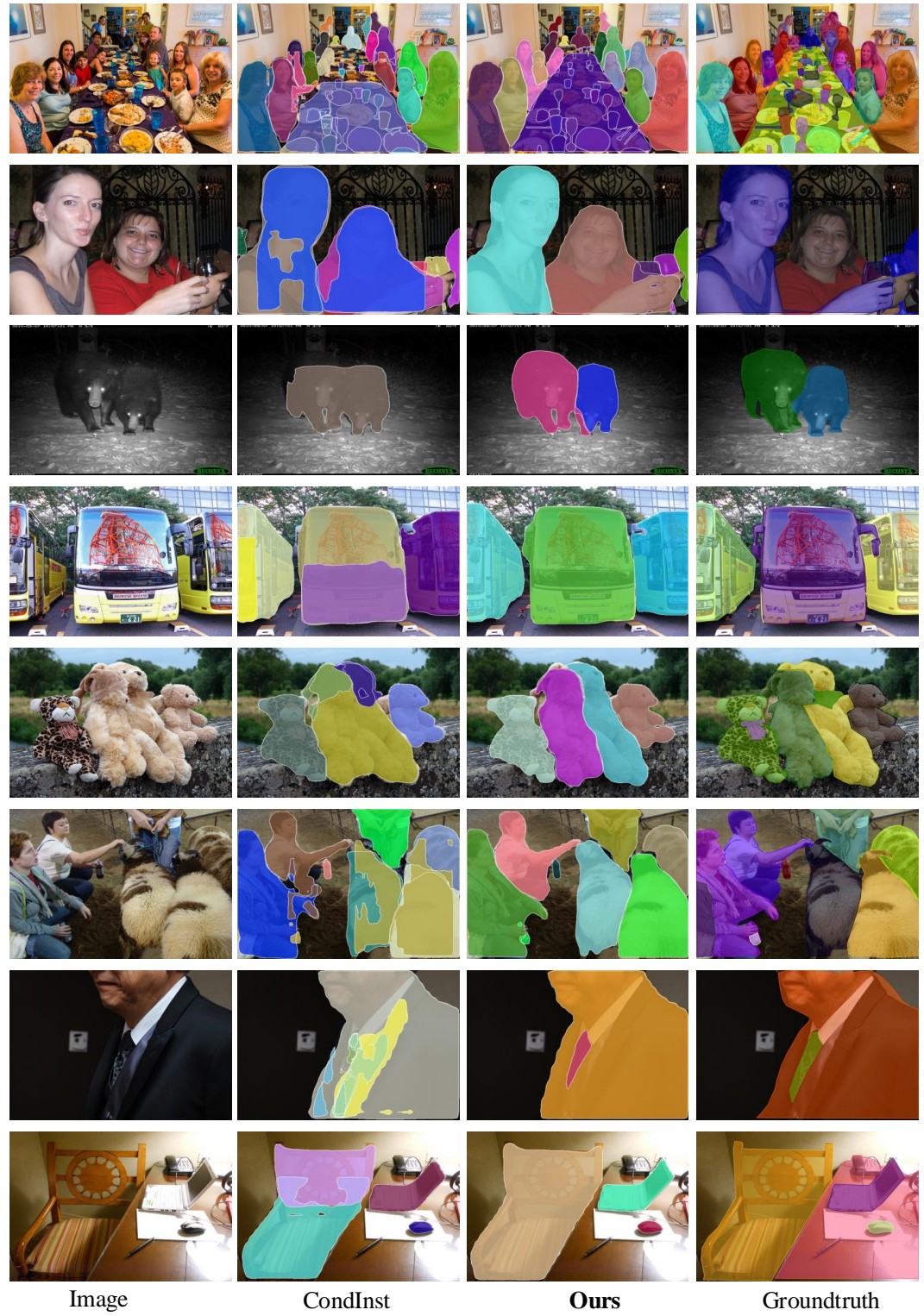

Image        CondInst        **Ours**        Groundtruth

Figure S1: **More qualitative results** on COCO [1] val2017 over CondInst [9]. See §S2 for details.

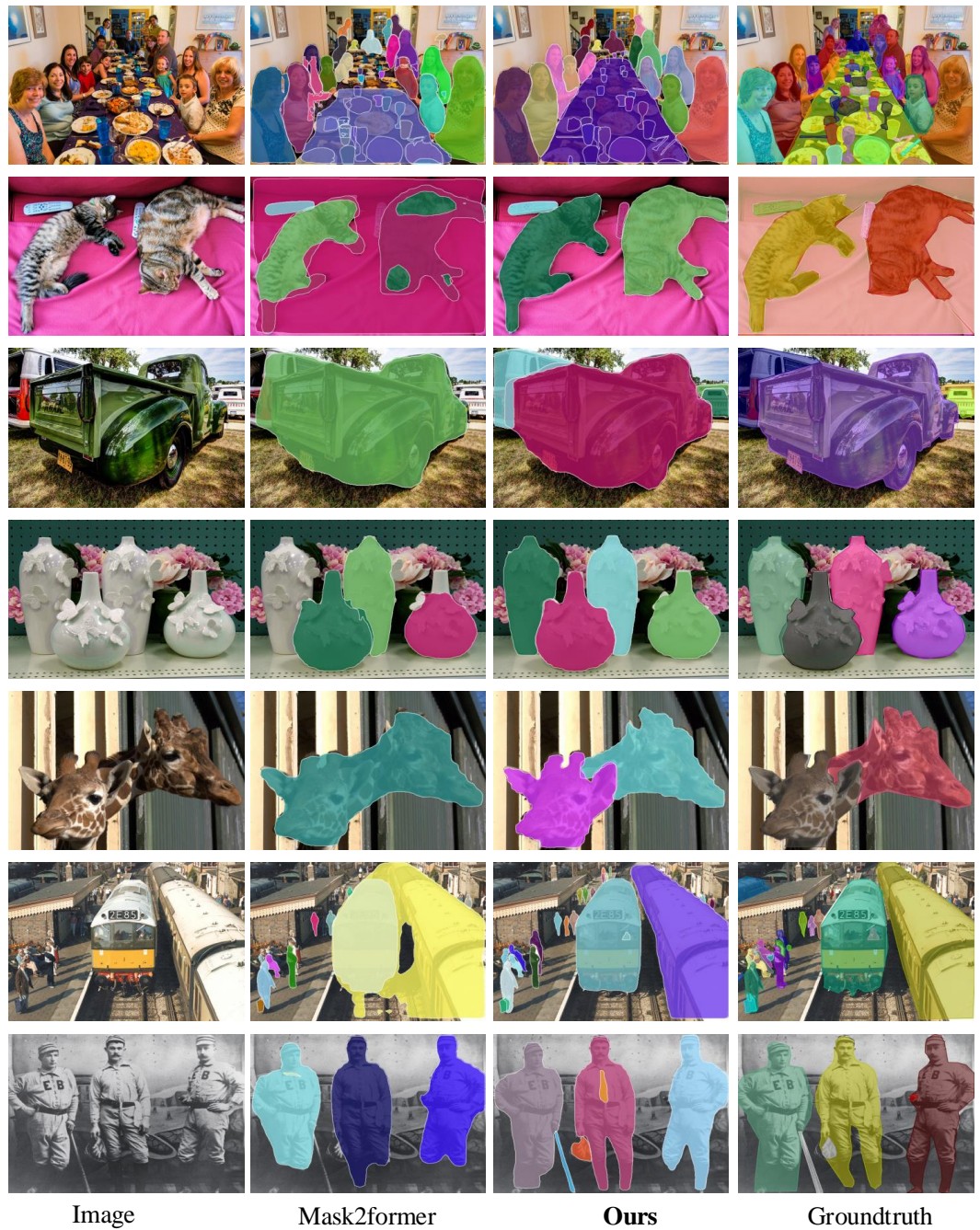

Image          Mask2former          **Ours**          Groundtruth

Figure S2: **More qualitative results** on COCO [1] `val2017` over Mask2Former [7]. See §S2 for details.

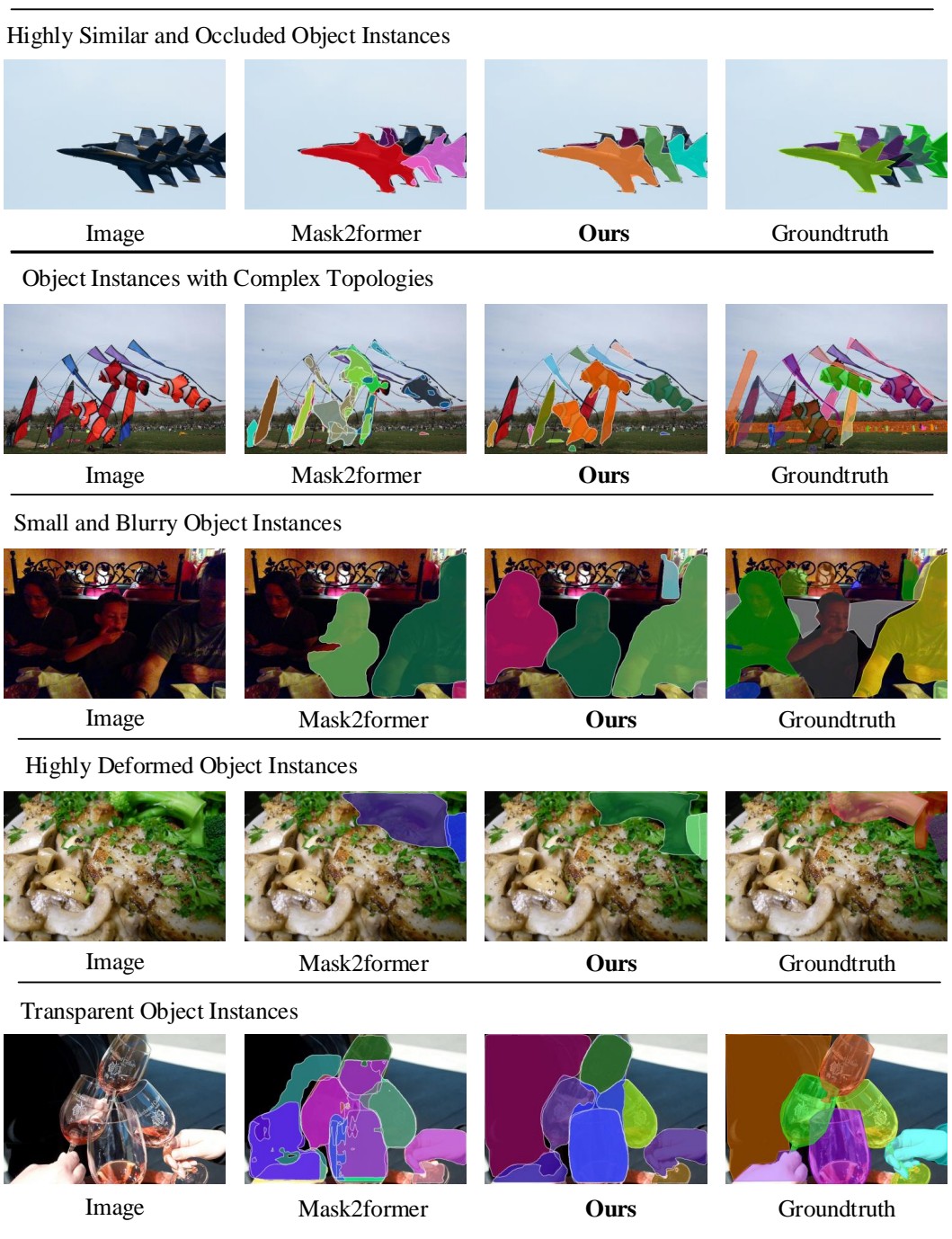

Figure S3: **Representative failure cases** on COCO[1] val2017. See §S2 for details.