# OpenReview forum: "Learning Equivariant Segmentation with Instance-Unique Querying"
_NeurIPS.cc/2022/Conference — NeurIPS 2022 Accept_

### Official Review · Reviewer_2dJn · 2022-07-10

**Rating:** 5
**Confidence:** 5
**Soundness:** 3 good
**Presentation:** 3 good
**Contribution:** 3 good

**Summary:**

- The paper proposes to enhance the dataset-level uniqueness and transformation equivariance of queries in the query-based segmentation methods.
- The proposed method essentially adopts some technics in contrastive learning to learn discriminative queries.
- The method consistently improves baseline methods like CondInst, SOLOv2, SOTA, and Mask2Former.

**Questions:**

1. Why color jittering and blur is not suitable? It is quite common in contrastive learning [58,59,60] to use both geometric and photometric transformations, therefore, intuitively the reviewer does not understand why it is not suitable.
2. Methods like MaskFormer, Mask2Former, and K-Net are not only used for instance segmentation but also panoptic segmentation methods, does the proposed method can have a more generic form that works for panoptic segmentation or it can only be used for instance discrimination?

**Limitations:**

1. As asked in the questions, if the methods only applies to instance segmentation, it might be better to claim its scope.

**Strengths And Weaknesses:**

## Strengths:
1. The paper exhibits consistent significant improvements over many methods including CondInst, SOLOv2, SOTR, and Mask2Former.
2. The paper reveals a new direction of improvements over query-based instance segmentation methods by enhance the discrimination ability of queries.

## Weaknesses:
1. There are too many equations that makes the paper hard to follow. For example, the notion of $O^I_{n}$ in eq.5 and  $\hat{W^{g}_{n}}$ in eq. 8
2. May also show the iteration time and GPU memory cost in Table 3 (b) and (c) to show the trade-off.
3. K-Net [a] is not mentioned and compared in the paper, which has better speed-accuracy trade-off than SOLOv2 and CondInst.
4. The description is not accurate and should be updated. This paper essentially improves methods that use dynamic kernels, i.e., ‘kernel-based’ methods rather than 'query-based' methods. SOLOv2 [37], CondInst[32], SOTR[39], K-Net [a], MaskFormer[40], Mask2Former [38] they essentially all use some strategy to predict content-aware kernels then use the kernels to perform convolution with the feature map to produce masks. This is explicitly described in the original paper of SOLOv2, CondInst, SOTR, and K-Net, although MaskFormer and Mask2Former describe them as ‘mask embeddings’.
5. The augmentation techniques are not sufficiently studied, i.e., only geometric transformation is studied in Table S2 while photometric transformations and the detailed parameters like cropping range are not studied.
[a] K-Net:Towards Unified Image Segmentation, NeurIPS 2021.

---

> ### Author Response · Authors · 2022-08-02
> **Response to Reviewer 2dJn**
>
> We thank the reviewer for the time and constructive feedback. We address the main questions below:
>
>
> #### **Q1. Too many equations and the notion of $\hat{O}^{I'}\_{n}$ and $\hat{W}^{g}\_{n}$.**
>
> **A1:** The definition of $\hat{O}^{I'}\_{n}$ and $\hat{W}^{g}\_{n}$ have been given in Line 176 and Line 240. We will try our best to improve our presentation and formulation.
>
> ---
>
> #### **Q2. Show the iteration time and GPU memory cost in Table 3 (b) and (c) to show the trade-off.**
>
> **A2:** For Table 3 (b), as the external memory capacity is fixed, there is no variation in training speed or GPU memory cost across different sampling strategies.
>
> For Table 3 (c), please find the updated version below:
>
> | capacity  |  AP |  AP$_{S}$ | AP$_{M}$  | AP$_{L}$  | Training speed (minutes/epoch)  | GPU memory cost (GB) |
> |:-:|:-:|:-:|:-:|:-:|:-:|:-:|
> |-  | 35.5  | 16.9  | 38.9  | 50.2  | 90  | 19.68  |
> | mini-batch  |  36.9 | 17.8  | 40.1  | 53.1  | 106  | 19.68  |
> | 10k pixels  | 37.0  | 17.9  | 40.3  | 53.2  | 91  | 20.24  |
> | 50k pixels  | 37.3  | 18.1  | 40.4  | 53.6  | 93  | 22.37  |
> | 70k pixels  | 37.5  | 18.5  | 40.4  | 53.7  | 93  | 23.72  |
> | **100k pixels**  | 37.6  | 18.6  | 40.6  | 54.0  | 94  | 25.37  |
>
> As seen, the optimal configuration of the memory capacity, *i.e.*, 100k pixels, only causes marginal training speed delay (~5%), as we mentioned in our manuscript.
>
> ---
>
> #### **Q3. K-Net.**
>
> **A3:** K-Net will be cited and involved in the comparison.
>
> ---
>
> #### **Q4. 'kernel-based' *vs* 'query-based'.**
>
> **A4:** Both the terms 'query-based' and 'kernel-based' are widely adopted terminologies in this field [18, 41, 42]. As the reviewer mentioned, MaskFormer and Mask2Former even use 'mask embeddings' (and we personally feel it is hard to say their mask embeddings are just dynamic kernels). We do not think this is a big issue. And our idea, clearly, is general for the query-matching-based instance segmentation paradigm.
>
> ---
>
> #### **Q5. Photometric transformation.**
>
> **A5:** This is due to the nature of transformation-equivariance. We have clearly discussed this issue in Line 68 in the suppl.. As our algorithm arises from the restriction of equivariant transformations that have to be elements from a group of linear transformation operators. Therefore, we can only adopt ordinary linear transformation (*i.e.*, flipping and cropping); arbitrary transformation (*i.e.*, color jittering and blur) is not applicable for our algorithm.
>
> Specifically, we encourage the transformation $g$ on feature representation to be ``equivariant against input imagery transformations'', which states:
>
> \begin{equation}
> \forall g\in G:  f(g(I))\approx g(f(I))=g(\textbf{I}).
> \end{equation}
>
> For color jittering and blurring, **no existing solution can obtain a non-trivial transformation $g$ to accommodate that the output representation changes in the exact same way with the same transformation $g$ applied to the input $I$**.
>
> But your question points out our ongoing study: How to find out such a corresponding non-linear homomorphic transformation $g'$?
>
> ---
>
> #### **Q6. Panoptic segmentation.**
>
> **A6:**  Yes, our method is generic for query-based models, irrespective of instance or panoptic segmentation. For panoptic segmentation, the only modification is for the stuff classes; as the stuff classes are not instance-discriminative, the training objectives for cross-scene querying (Eq. 5) of stuff classes should be the groundtruth stuff masks, instead of all-zero matrices.
>
> Below we further report additional experimental results on MS COCO Panoptic, on the top of Mask2Former. We can find that our algorithm improves PQ by **1.2**.
>
> | Method  | Backbone  | #Epoch  | PQ  | PQ$^{th}$ | PQ$^{st}$ |
> | :-: | :-: | :-: | :-: | :-: | :-: |
> | Mask2Former |  Swin-B | 50  | 56.1  | 62.5  | 46.7  |
> | **+Ours** |  Swin-B | 50  | **57.3**  | **64.1**  | **47.4**  |

---

### Official Review · Reviewer_XmTr · 2022-07-11

**Rating:** 5
**Confidence:** 4
**Soundness:** 2 fair
**Presentation:** 3 good
**Contribution:** 3 good

**Summary:**

This paper proposes using two ways to improve the performance of query-based instance segmentation network. First, by using the queries to retrieve the corresponding instance from the whole training dataset, the segmenters are forced to learn instance-uniques queries. Second, by performing geometric transformations and encouraging the network predictions to be equivariant, the authors expect to learn more robust instance-query matching. Experiments show that both ways can improve the instance segmentation effectively.

**Questions:**

See the weakness part.

**Limitations:**

The authors didn't discuss the limitations and potential negative societal impact of their work.

**Strengths And Weaknesses:**

Pros:
-The idea is simple and reasonable.

-Overall, the paper is easy to read and understand.

-Experiment results are good.

Cons:
-The relationship between the proposed two contributions seems not to be clear. And the equivariance-based augmentation seems not to be tailored for instance segmentation, and even not to be tailored for query-based segmentation framework. I think it is a common augmentation trick which may not be suitable to claim it as a contribution of this paper.

-What is the detailed format of cross-entropy loss and focal loss for $\mathcal{L}_{inter\_mask}$？

---

> ### Author Response · Authors · 2022-08-02
> **Response to Reviewer XmTr**
>
> We thank the reviewer for the time, and constructive feedback. We address the main questions below:
>
> #### **Q1. Relationship between Uniqueness and Equivariance.**
>
> **A1:** As we repeatedly mentioned in our manuscript, our contributions: dataset-level uniqueness and transformation equivariance, are orthogonal and respectively address two core properties of the query-based instance segmentation paradigm (Line 3, Line 39, Line 88, Line 158). Neither of them has been explored so far. And we empirically show that, uniqueness and equivariance both boost the performance (Line 330-335); as they are orthogonal, their integration achieves further better performance (Line 335-337).
>
> ---
>
> #### **Q2. Equivariance-based augmentation.**
>
> **A2:** Sorry for this confusion.
>
> First, please note that our approach is not an equivariance-based augmentation technique but an equivariant representation learning strategy for any query-based segmenter. Concretely, traditional data augmentation methods only deal with input data transformation (input images and labels), with no constraint about the relation between the feature representations and queries produced from the transformed views. They just simply use the transformed images and annotations as **additional individual training examples**. In contrast, our approach establishes the valuable equivariance property of both query embedding and feature representation with respect to transformations (Line 109), which benefits per-instance description for mask generation.
>
> Second, the key is NOT the input transformation; the key is the transformation equivariance property. We enforce the matching between the query and feature to be equivariant against input imagery transformations. This view is fresh and insightful.
>
> Third, traditional augmentation strategy in query-based instance segmentation can be viewed as a special case of our approach, *i.e.* Eq. (7). Please refer to Line 231-233 for more detailed discussion.
>
> Forth, we provide extensive comparison with traditional augmentation strategy. In Table 3(d), the second row refers to the performance of traditional augmentation strategy. In Table S2 in the suppl., we provide very detailed comparison. These experimental results clearly and comprehensively demonstrate the advantage of our equivariance learning strategy over the common data augmentation technique.
>
> ---
>
> #### **Q3. Format of cross-entropy loss and focal loss for $\mathcal{L}_{inter\\_mask}$.**
>
> **A3:** Cross-entropy loss and focal loss, in our case, are in the formats of:
>
> \begin{equation}
>     \frac{1}{H \times W}\sum^{H \times W}\_{i}-log(1-\hat{O}^{I'}\_{n,\ i})
> 	\quad
> 	\text{and}
> 	\quad
>     \frac{1}{H \times W}\sum^{H \times W}\_{i}-(\hat{O}^{I'}\_{n,\ i})^{\gamma} log(1-\hat{O}^{I'}\_{n,\ i})
> \end{equation}
>
> where $\hat{O}^{I'}\_{n}\in[0, 1]^{H \times W}$ refers to the $n$-*th* inter-image instance prediction mask for image $I'$ (Line 173-179), and $\hat{O}^{I'}\_{n,\ i}\in\hat{O}^{I'}\_{n}$.
>
> ---
>
> #### **Q4. Limitation and social impact.**
>
> **A4:** We clarify we have discussed the limitation and broad impact in S4 in suppl..

---

### Official Review · Reviewer_jN1s · 2022-07-12

**Rating:** 5
**Confidence:** 4
**Soundness:** 3 good
**Presentation:** 3 good
**Contribution:** 3 good

**Summary:**

This paper proposes a new training framework that exploits query embedding learning. Dataset-level uniqueness and transformation equivariance are introduced and demonstrate promising results on the benchmark datasets.

**Questions:**

In Ln 158, uniqueness and robustness are named as two crucial properties. Can these be consistent with the statement in abstract Ln 5?

If the equivariance is estimated on the feature representation, what makes it different to traditional concepts, e.g. invariance?

**Limitations:**

Despite the novelty of introducing equivariance, the in-depth analysis of the mathematical properties is limited. Only empirical results cannot fully reflect the rationale behind the equivariance eqution.

**Strengths And Weaknesses:**

+ Framework contribution: this work brings a new paradigm shift that goes beyond inner-scene training to an inter-scene level query embedding separation.

+ Introducing equivariance is intuitive and the results are convincing.

- The evaluation is somewhat weak. Only COCO is adopted.

---

> ### Author Response · Authors · 2022-08-02
> **Response to Reviewer jN1s**
>
> We thank the reviewer for the time and constructive feedback. We address the main questions below.
>
> #### **Q1. Uniqueness and robustness.**
>
> **A1:** Thanks for your careful review. Yes, here dataset-level uniqueness and transformation equivariance (Line 5) are referred to simply as ''Uniqueness'' and ''Robustness''; or, in other words, the desired ''Uniqueness'' and ''Robustness'' properties are achieved by addressing ''dataset-level uniqueness'' and ''transformation equivariance'' (Line 158). After reading your comments, we feel such statement may cause some misleading. We will rephrase related sentences.
>
> ---
>
> #### **Q2. If the equivariance is estimated on the feature representation, what makes it different to traditional concepts, *e.g.* invariance?**
>
> **A2:** Sorry for this confusion. As we mentioned in Line 101-103, invariance is a special case of equivariance. For invariance, the feature representation is desired to NOT vary with the input transformation. That is to say, given a representation $f$ and a transformation $g$ for input $I$, invariance can be expressed as: $f(g(I)) \approx f(I)$. Differently, the representation $f$ is said to be equivariant with $g$ if $f(g(I)) \approx g(f(I))$.
>
> Invariance is not suitable for instance segmentation task, as the input transformation should cause an exact change (instead of no change) in the segmentation mask and feature map. This is also why we address equivariance -- it essentially addresses the very nature of this task.
>
> ---
>
> #### **Q3. The evaluation is somewhat weak. Only COCO is adopted.**
>
> **A3:** We clarify that we follow the standard evaluation protocol in this field and test our algorithm on the top of SIX famous instance segmentation models, *i.e.*, CondInst, SOLOv2, SOTR, Mask2Former (in the main paper), SparseInst and SOLQ (in the supplementary), and different backbone networks, *i.e.*, ResNet-50, ResNet-101, and Swin-S/B/L.
>
>
> To better address your concern, we conduct additional experiments on LVISv1 [ref1]. The results are listed below. On top of SOLOv2, our algorithm provides significant performance gain, *i.e.*, **2.7**  mask mAP. Our experimental results on COCO and LVISv1 thoroughly demonstrate the power of our idea and the effectiveness of our algorithm.
>
>
> | Method | Backbone | #Epoch  | AP  |  AP$_{50}$ |AP$_{75}$ |AP$_{S}$ |AP$_{M}$ |AP$_{L}$ |AP$_{r}$ |AP$_{c}$ |AP$_{f}$ |
> | :-: | :-: | :-: | :-: | :-: | :-: | :-: | :-: | :-: | :-: | :-: | :-: |
> | SOLOv2  | Resnet-50  |  36 | 21.4  | 34.0  | 22.8  | 14.9  | 29.1  | 34.8  | 9.5 | 20.9 | 27.6 |
> | **+Ours** | Resnet-50  |  36 | **24.1**  | **37.4**  | **25.5**  | **17.0**  | **31.5**  | **39.1**  | **13.5** | **22.8** | **29.7**
>
>
> In addition, our method is even generic for query-based models, irrespective of instance or panoptic segmentation. For panoptic segmentation, the only modification is for the stuff classes; as the stuff classes are not instance-discriminative, the training objectives for cross-scene querying (Eq. 5) of stuff classes should be the groundtruth stuff masks, instead of all-zero matrices.
>
> Below we further report additional experimental results on MS COCO Panoptic, on the top of Mask2Former. We can find that our algorithm improves PQ by **1.2**.
>
> | Method  | Backbone  | #Epoch  | PQ  | PQ$^{th}$ | PQ$^{st}$ |
> | :-: | :-: | :-: | :-: | :-: | :-: |
> | Mask2Former |  Swin-B | 50  | 56.1  | 62.5  | 46.7  |
> | **+Ours** |  Swin-B | 50  | **57.3**  | **64.1**  | **47.4**  |
>
>
> [ref1] LVIS: A Dataset for Large Vocabulary Instance Segmentation

---

### Official Review · Reviewer_Zz8f · 2022-07-13

**Rating:** 7
**Confidence:** 3
**Soundness:** 3 good
**Presentation:** 4 excellent
**Contribution:** 3 good

**Summary:**

This work proposes a novel instance segmentation training paradigm that can be combined with any model. Two properties are explored: dataset-level uniqueness and transformation equivariance. This training is combined with quite a few SOTA models, and it is shown that a significant gain of 2-3 AP on COCO. Training is only 5% slower than standard training, with no speed loss at instance.

**Questions:**

I asked my questions in weaknesses. Authors should focus on those in the rebuttal.

**Limitations:**

Nothing to add here.

**Strengths And Weaknesses:**

Strengths:
+ This training paradigm is novel as far as I can tell, it is properly motivated, clearly explained and both additions are orthogonal, which results in experiment section clearly show
+ When I started reading the paper, I expected that training cost will be much higher than baseline, but upon reading the implementation strategies (namely external memory, sparse sampling and instance-balanced sampling) I was convinced that the loss in speed will be minor, and it was, it is just 5%! Kudos for the engineering aspect of the paper
+ Experimental section is very well designed. Authors compare with a lot of SOTA approaches, they incorporate their method with 4 different representative models. Then, they clearly explain the effect of inter-scene and equivariance, they are both providing performance boost and are orthogonal, so combined they achieve biggest jump. Finally, ablation over hyperparams is very good, it shows that the newly added hyper-parameters are fairly robust (at least on COCO dataset), and it helps the reader select them on their problem.
+ Performance gains are very good, hard to be ignored. Performance of the best result is SOTA.

Weaknesses:
- Whole paper evaluated on only one dataset. I do know COCO is standard, but for a paper that claims to improve training paradigm, I think maybe one more dataset should be used. Not to achieve SOTA necessarily, but to show benefits on top of a good model. Eg, evaluation on LVIS would greatly improve the paper, in my opinion.
- Minor: When reading Mask2Former paper, it seems their equivalent COCO result is 50.1, not 48.5 like here. Can authors explain in more detail what is the difference in their implementation and why this happens?

---

> ### Author Response · Authors · 2022-08-02
> **Response to Reviewer Zz8f**
>
> We thank the reviewer for the time and constructive feedback. We address the main questions below.
>
> #### **Q1. Show benefits on top of a good model. Eg, evaluation on LVIS.**
>
> **A1:** To address your concern, we conduct experiments on LVISv1. The results are listed below. On top of SOLOv2, our algorithm provides significant performance gain, *i.e.*, **2.7** mask mAP. Our experimental results on COCO and LVISv1 thoroughly demonstrate the power of our idea and the effectiveness of our algorithm.
>
>
> | Method | Backbone | #Epoch  | AP  |  AP$_{50}$ |AP$_{75}$ |AP$_{S}$ |AP$_{M}$ |AP$_{L}$ |AP$_{r}$ |AP$_{c}$ |AP$_{f}$ |
> | :-: | :-: | :-: | :-: | :-: | :-: | :-: | :-: | :-: | :-: | :-: | :-: |
> | SOLOv2  | Resnet-50  |  36 | 21.4  | 34.0  | 22.8  | 14.9  | 29.1  | 34.8  | 9.5 | 20.9 | 27.6 |
> | **+Ours** | Resnet-50  |  36 | **24.1**  | **37.4**  | **25.5**  | **17.0**  | **31.5**  | **39.1**  | **13.5** | **22.8** | **29.7**
>
>
> In addition, our method is even generic for query-based models, irrespective of instance or panoptic segmentation. For panoptic segmentation, the only modification is for the stuff classes; as the stuff classes are not instance-discriminative, the training objectives for cross-scene querying (Eq. 5) of stuff classes should be the groundtruth stuff masks, instead of all-zero matrices.
>
> Below we further report additional experimental results on MS COCO Panoptic, on the top of Mask2Former. We can find that our algorithm improves PQ by **1.2**.
>
> | Method  | Backbone  | #Epoch  | PQ  | PQ$^{th}$ | PQ$^{st}$ |
> | :-: | :-: | :-: | :-: | :-: | :-: |
> | Mask2Former |  Swin-B | 50  | 56.1  | 62.5  | 46.7  |
> | **+Ours** |  Swin-B | 50  | **57.3**   | **64.1**  | **47.4**  |
>
> ---
>
> #### **Q2. Mask2Former' results on COCO.**
>
> **A2:** Thank you so much for your careful review. Our previous re-implementation was based on mmdetection, which only released code for panoptic segmentation at the time of submission. When we modified and adopted the mmdetection version of Mask2Former to COCO instance segmentation, some hyperparameters were not changed and we actually ran fewer iterations compared with the original Mask2Former. With your reminder, we rechecked our code and solved this issue. Below are the new results. As seen, the scores of our reproduced Mask2Former are almost the same as the original ones, and our algorithm boosts the performance by **1.6** mask mAP.
>
> | Method  | Backbone  | #Epoch  | AP  |  AP$\_{50}$ |AP$\_{75}$ |AP$\_{S}$ |AP$\_{M}$ |AP$\_{L}$ |
> | :-: | :-: | :-: | :-: | :-: | :-: | :-: | :-: | :-: |
> | Mask2Former |  Swin-L | 100  | 50.2  | 74.8  | 54.7  | 29.2  | 53.8  | 71.1  |
> | **+Ours** |  Swin-L | 100  | **51.8**  | **76.0**  | **56.8**  | **29.9**  | **55.1**  | **73.3**  |

---

> > ### Comment · Reviewer_Zz8f · 2022-08-03
> > **Response to Rebuttal**
> >
> > I have read the response to my review and all other reviews, and I think authors covered the issues pretty well. I do believe adding the new results on the additional datasets will make the paper stronger. The newly proposed training paradigm is novel, and experiments support consistent significant improvements over popular models. Hence, I stay by my initial rating and propose to accept the paper.

---

### Meta-Review · Area_Chair_jhfY · 2022-08-24

**Recommendation:** Accept
**Confidence:** Certain

**Metareview:**

This paper leverages dataset-level uniqueness and transformation equivariance to improve state-of-the-art instance segmentation methods.
The reviews were overall positive about the submission: the reviewers especially highlighted the good experimental results, the relevance of the scene level query embedding and its complementarity with the equivariance constraints.
The authors' feedback brings important answers to some reviewers' concerns. Especially, the new conclusive experiments on LVIS or the extension of the method for panoptic segmentation widens the approach's applicability and has been appreciated. Other answers in the rebuttal did not convinced some reviewers, and there remains issues about the novelty of the approach, the terminology and positioning with respect to 'query-based' approaches, or the extension of the method for photometric equivariance.

The AC carefully read the submission. The AC considers that the idea of querying instances from the whole training dataset is interesting. Despite the limited contribution of the equivariance loss, which has been used in several related scenarios, the design of the whole approach in the context of instance segmentation is relevant and well designed. The experiments are also convincing. It is a pity that the authors did not take the opportunity to update the paper during the discussion period, especially the new experiments and the clarifications requested by the reviewers. Based on the relevance of the approach and its good experimental results obtained over various baselines in several datasets, the AC recommends acceptance. He highly encourages the authors to include the elements discussed in the rebuttal to improve the quality of the final paper.


**Award:**

No

---

### Decision · Program_Chairs · 2022-09-14

Accept